# Bacterial Isolates Derived from Nest Soil Affect the Attraction and Digging Behavior of Workers of the Red Imported Fire Ant, *Solenopsis invicta* Buren

**DOI:** 10.3390/insects13050444

**Published:** 2022-05-07

**Authors:** Nicholas V. Travanty, Edward L. Vargo, Coby Schal, Charles S. Apperson, Loganathan Ponnusamy

**Affiliations:** 1Department of Entomology and Plant Pathology, North Carolina State University, Raleigh, NC 27695, USA; travanty@gmail.com (N.V.T.); coby@ncsu.edu (C.S.); apperson@ncsu.edu (C.S.A.); 2Department of Entomology, Texas A&M University, College Station, TX 77840, USA; ed.vargo@tamu.edu; 3Comparative Medicine Institute, North Carolina State University, Raleigh, NC 27695, USA

**Keywords:** *Solenopsis invicta*, bacterial isolates, behavioral responses, digging behavior

## Abstract

**Simple Summary:**

Populations of the red imported fire ant (*Solenopsis invicta*) are found throughout the southern United States. Because these invasive ants sting and are highly territorial, they are hazardous to people and livestock and are detrimental to native ant populations. Control of this species generally relies on insecticidal baits that attract and kill the ant. The aim of our study was to determine if bacteria cultured from *S. invicta* nest soils affected worker ant behaviors and whether the bacteria were attractive or repellent to the ants. Bacterial isolates cultured from nest soils were used in binary choice bioassays that tested for effects of bacterial species and bacterial concentrations on worker ant digging and residing preferences. *Arthrobacter woluwensis* (Actinobacteria) attracted worker ants while bacteria identified as Firmicutes generally repelled ants. This study provides a basis for the identification of new biologically derived compounds that can be used to alter behaviors of the red imported fire ant and be implemented in novel control strategies.

**Abstract:**

Populations of monogyne and polygyne red imported fire ants (RIFA), *Solenopsis invicta* Buren, are distributed throughout the southern United States. This ant species is hazardous to farm animals and workers, damages infrastructure, and depletes native arthropod populations. Colony expansion is affected by several biotic factors, but the effects of soil microbes on ant behavior related to soil excavation within nest sites have not been investigated. Consequently, we cultured bacteria from RIFA nest soils. The effects of individual bacterial isolates and bacterial cell densities on the choice of digging site as well as digging activity of monogyne and polygyne RIFA worker ants were evaluated in two-choice bioassays. Based on phylogenetic analysis, 17 isolates were selected and tested initially at 5 × 10^8^ cells/mL and 20 workers per assay. Firmicutes (*Bacillus*, *Paenibacillus*, *Brevibacillus*) repelled the ants, but *Arthrobacter woluwensis* strongly attracted ants. Subsequently, the six isolates having the greatest positive or negative effects on ant behavior were evaluated at a lower bacterial cell and worker ant densities. Ant responses to these bacteria generally decreased as cell densities declined to 5 × 10^6^ cells/mL. Observations of ant behavior during a three-hour, two-choice bioassay revealed that ants generally visited both control and bacteria-treated sand prior to making a digging site choice. Our research results indicate that soil bacteria may mediate ant nest expansion or relocation and foraging tunnel construction. Identification of bacterial metabolites that affect RIFA digging behavior merits additional research because these compounds may provide a basis for novel management strategies that repel RIFA away from sensitive infrastructure or attract fire ants to insecticidal baits.

## 1. Introduction

The red imported fire ant (RIFA), *Solenopsis invicta* Buren, is suspected to have arrived in the United States during the 1930s to 1940s through the port of Mobile, Alabama [1]. This species has since increased its range and now inhabits rural and urban areas throughout the southern United States. RIFAs have also been dispersed to locations in the Caribbean, Asia, and Oceania [1,2,3]. This species lives in subterranean colonies that include tens of thousands of worker ants. *Solenopsis invicta* may out-compete native ant species for resources [4], a behavior that likely facilitated its successful invasion of North America. Each worker ant in the colony is armed with a stinger used for prey acquisition and colony defense. RIFAs invade open areas and deliver venom through their stingers, making them a serious pest to livestock and agricultural workers [5]. They also negatively affect irrigation systems, electrical and mechanical equipment, some ground-dwelling vertebrates, and crop production through predation on beneficial arthropods [6,7,8]. 

The social organization of *S. invicta* includes both single-queen (monogyne) and multiple-queen (polygyne) colonies. The social structure of a RIFA colony is determined by the composition of a supergene present on the “social chromosome.” Monogyne queens and workers have two copies of the SB form of the social chromosome, whereas in polygyne colonies, queens are SB/Sb and workers are either Sb/SB or SB/SB [9]. Monogyne workers are typically characterized as having greater individual body mass and are more aggressive in colony defense and in resource competitiveness than polygyne workers. Monogyne colonies are often composed of a single large mound, whereas several small interconnected mounds may constitute a single polygyne colony [10].

RIFA colony founding typically occurs on open, disturbed terrain [11]; colonies are common on roadsides [12], disturbed forests [13], and in pastures [14]. Several abiotic and biotic factors contribute to RIFA nest site selection. RIFA nests are typically located in areas provided with direct sunlight [15]. The distribution of the species is limited by freezing temperatures [16], which limits the northward expansion of North American populations. Arid conditions are also a limitation of RIFA colony development [16]. Biotic factors such as vegetation [17] and the presence of potentially competitive ants [11] are deterrents to colony founding. Soil microorganisms, such as entomopathogenic fungi, also impede RIFA colony development [18]. 

At present, the main methods used to control and prevent the spread of RIFA include chemical insecticides, physical disruption of ant mounds, plant quarantine regulations, and biological control [19]. Among these methods, synthetic insecticides are most frequently used for the control of fire ants through broadcast applications or treatment of individual mounds with toxic baits or contact insecticides. Biological control methods have been used on a limited basis to control *S. invicta* [20,21]. The RIFA has been reported to be naturally infected with pathogenic microorganisms, including *Aspergillus* spp. [22,23], *Tetradonema solenopsis* [24], *Beauveria bassiana* [25], and *Thelohania solenopsae* [26]. However, environmental conditions that promote epizootics are not well understood, and utilizing these microorganisms for the control of *S. invicta* populations has been problematic. Classical biocontrol programs have been developed using *Pseudacteon* flies. Biocontrol using these parasitoids of *S. invicta* has been demonstrated to be an effective control strategy when incorporated into an integrated pest management program [5].

Microbial volatiles have been shown to significantly affect insect behavior; in particular, these semiochemicals mediate interactions between bacteria and insects [27]. Bacterial metabolism may be functionally interconnected with some insects to produce semiochemicals that affect specific insect populations and their social behavior [28,29]. Furthermore, semiochemicals have been utilized in several ways to control insect pest populations, including for monitoring and detection, population suppression through mass trapping, and in attract-and-kill techniques [27,30]. Volatile semiochemicals of soil microbes have been demonstrated to affect nesting site preferences of RIFAs [31]. Specifically, Huang et al. [31] demonstrated that RIFA queens and workers were attracted to nest-associated soil that contained the Actinobacteria *Streptomyces* and *Nocardiopsis*, which produced the volatiles geosmin and 2-methylisoborneol. Nevertheless, individual bacterial species from the broader soil bacterial community have not been evaluated for their effects on RIFA behavior, and differences in behavioral responses to bacteria between RIFA social forms have not been studied. Because fire ants are edaphic insects, it is possible that soil bacteria affect the behavior of RIFAs by influencing the choice of digging site by workers and eliciting a digging response. RIFAs may be attracted or repelled by soil-derived bacteria depending on the bacterial species and their concentrations. The aim of the present study was to investigate how cultivable soil-derived bacteria influence digging site preference by fire ant workers and affect their digging activity. Accordingly, we cultured, isolated, and identified bacterial species from RIFA nest soils. Using a two-choice digging assay, we characterized the behavioral responses of worker ants to selected bacterial species over a range of cell densities. Furthermore, we included both monogyne and polygyne worker ants in our assays to determine whether the two social forms exhibited the same behavioral responses to bacteria.

## 2. Materials and Methods

### 2.1. Soil Collection and Isolation of Bacteria from RIFA Nests 

Using previously described sterile collection methods [32], soil and worker ants were collected from eight RIFA nests (four monogyne and four polygyne). Briefly, a sterilized steel corer was inserted into nest tumulus, from which soil samples (approx. 5 g) were taken from 0, 10, and 20 cm depths and then separately placed into sterile polycarbonate centrifuge tubes. Workers were collected by allowing ants to walk onto a wooden rod and transferring them into sterile polycarbonate centrifuge tubes. Each nest was located within approx. 800 m radius of the Dearstyne Entomology Building (35.78876° N, −78.69913° W) on the North Carolina State University campus; collections were performed in October 2015. Ant social form was verified by genotyping assays using the *Gp-9* locus [33], a locus associated with the social chromosome. Within one hour of soil collection, each soil sample (1 g) was sifted through a sterilized steel mesh screen (1 mm pore size), placed into a sterile 50 mL polypropylene conical centrifuge tube (cat. 430290; Corning Falcon, Tewksbury, MA, USA) containing 25 mL of sterile saline solution (0.85 % NaCl), capped and shaken (100 rpm, 60 min). Each such bacterial suspension was serially diluted in sterile saline and spread-plated on trypticase soy agar (TSA; Difco, Detroit, MI, USA), incubated at 28 °C, and monitored for colony growth for 96 h. To confirm that cultivable bacteria detected in culture assays originated from soil samples and were not a product of environmental contamination during processing, a 1 g sample of autoclaved sand was also subjected to sifting, suspension, dilution, and culturing using the same procedures as the field-collected soil samples. Morphologically distinct colonies were picked and subcultured by streaking onto the same medium several times to obtain pure isolates. The bacterial isolates were stored individually in glycerol at −80 °C.

### 2.2. Identification of Bacterial Isolates

A single colony from each bacterial isolate was suspended in sterile water (25 µL), boiled for 10 min, and centrifuged. We used a 2-µL portion of the supernatant as a PCR template to amplify a 16S ribosomal RNA gene fragment using 27F and 1494R universal primers [34]. PCR conditions used were described by Ponnusamy et al. [35]. Amplification and expected amplicon size were confirmed by gel electrophoresis. Amplicons were directly Sanger-sequenced using 27F primers (Eton Bioscience, Research Triangle, NC, USA). The resulting sequences were aligned by MEGA ver. 7.0.18 [36] using the ClustalW algorithm [37] and trimmed to 600 bases. To identify redundant isolates, a Jukes–Cantor distance matrix was created between all pairs of bacterial isolates. Sample clusters were identified, and a representative bacterial species from each cluster was randomly selected to be used in RIFA behavioral bioassays (described below). Using the maximum likelihood method (Tamura-Nei model) with bootstrapping at 500 replications [38], a phylogenetic tree was constructed from the selected bacteria. Bacterial phylotypes were assigned using the NCBI Database (accessed 8 November 2020) using the BLASTn algorithm [39] and Ribosomal RNA Database [40].

### 2.3. Isolate Preparations

To measure the influence of bacterial isolates (Table 1) on ant behavior, 17 bacterial isolates were tested individually using the two-choice digging assay developed by Chen and Allen [41]. Sterile trypticase soy broth (TSB) (25 mL) held in sterile centrifuge tubes were each inoculated with a different bacterial isolate (10 µL) from glycerol stocks and incubated on an orbital shaker (28 °C, 200 RPM, 22 h). Following incubation, bacterial densities were measured with a hemocytometer (Fisher Scientific, Waltham, MA, USA). Each isolate was then centrifuged (7840× *g*, 10 min) into a pellet, the supernatant was discarded, and the bacteria were re-suspended in sterile 0.85% NaCl to bring the bacterial suspensions to approximate target densities of 5 × 10^8^, 5 × 10^7^, or 5 × 10^6^ cells/mL. For a control treatment, sterile 25 mL TSB medium was also prepared in the same manner but without an inoculum. After centrifugation, the supernatant was discarded, 10 mL sterile 0.85% NaCl was added to each tube, and the resulting saline solution was used as a control in behavioral bioassays (described below).

### 2.4. Collection and Maintenance of RIFA Colonies

From the same eight RIFA colonies (four monogyne, four polygyne) used for bacterial culturing, worker ants and nest soil (approx. 3.8 L from each nest) were collected and held in 11.4-L plastic buckets, the walls of which were coated with Fluon (cat. 2871D; BioQuip, Rancho Dominguez, CA, USA) to prevent ants from escaping. Sucrose solution (10% in DW), water, and crickets were provided to ants on a daily basis while they were held in the lab at 22 °C and 60% relative humidity on a 14:10 (L:D) h cycle.

### 2.5. Bioassay Setup and Validation

Bioassays were adapted from the *S. invicta* two-choice digging assay developed by Chen and Allen [41]. We modified this bioassay to include sand (Quikrete Premium, Home Depot, Cary, NC, USA) that was treated with a suspension of a bacterial isolate. Briefly, a plastic 100 mm dia. Petri dish (Fisherbrand, Thermo Fisher, Waltham, MA, USA) was used as a 2-choice arena (See Figure 1 for details). On the Petri dish bottom, the snap lids of two 1.25 mL centrifuge tubes were attached directly opposite and spaced 66 mm from one another. A hole (3 mm dia.) was drilled through the Petri dish bottom and each tube lid giving ants access to the interior of treatment and control sand tubes (described below) which were snapped onto the Petri apparatus for each assay. Worker ants (20 or 6 RIFA workers/assay, dependent on assay) were introduced into the assay arena and allowed access to the tubes to excavate sand. The quantity of sand excavated from each tube and the location of ants at the end of the assay were determined as outcomes (dependent variables). Prior to setting up each bioassay, sand was sifted through a 2 mm mesh sieve to remove large particles, autoclaved, and dried in a drying oven (50 °C, 24 h). Portions of the dry sand were transferred to sterile 1.25 mL centrifuge tubes. Each tube was filled to the top, capped, and weighed to the nearest mg using a precision scale (model XS-64, Mettler-Toledo, Columbus, OH, USA). Experimental tubes were treated with bacterial isolate suspensions and control tubes were treated with control preparations (previously described). Because 350 µL volumes of bacterial isolate suspensions were found to readily absorb into and throughout the dry sand without overflowing or pooling at the surface, this volume was used in all treatments.

Assays were performed in the same room and under the same environmental conditions in which the ants were kept. Assay arenas were located on a shelf with fluorescent lighting positioned overhead (approx. 40 cm above). To control for potential side bias of worker ant subjects, the orientation of the treatment and control sides of the assay dishes were alternated for each experimental trial.

The initial assays that each included 20 workers were terminated at 18 h immediately after which the number of ants residing in control sand tubes, treatment sand tubes, and other areas of the arena was recorded. Ants that were inside of sand tubes, positioned at the tube egress, or actively excavating sand from the tube were considered to have made a choice of the bacteria-treated or control sand tube. Ants in any other positions in the assay arena were determined to be in neutral areas and were scored as non-responders. The percentage of responders was calculated from the total number of ants introduced into the arena for each assay trial, and only responding ants were included in ant position analysis. Tubes containing the remaining sand were then carefully removed, dried, and the mass of each was measured so that the quantity of displaced sand could be recorded as a measure of digging activity.

During the assays, worker ant corpses were occasionally found in the neutral areas, and following the assays, some ants remained in the sand tubes and perished when subjected to the sand drying procedures. Because it could not be determined whether mortality of the ants remaining in the tubes occurred during or following the assays, mortality was not measured as an experimental outcome.

Because polygyne RIFAs are smaller than monogyne RIFAs, polygyne ants would likely excavate less soil per worker. For each assay, a digging preference index (DPI) [42] was also calculated to standardize digging activity of monogyne and polygyne ants. This index normalized displaced quantities by dividing the weight of sand removed from the test vial (after subtracting the amount of sand removed from the control vial) by the total weight of sand removed from the treatment and control vials. Index values ranged from −1.0 (indicating relative preference for control sand) to 1.0 (indicating relative preference for treated sand).

During the initial assays, it was observed that small quantities of moistened sand that had not been excavated adhered to the Petri dishes after sand tubes were removed and small quantities of sand occasionally spilled while capping/uncapping sand tubes. To ensure that these occasional sources of error did not bias bioassay outcomes, dummy assays (*n* = 32) without worker ants were set up and treated in the same manner as those exposed to worker ants to measure these errors. The average (±SE) amount of sand lost in dummy assays (error control sand + error treatment sand) was 7.9 (±1.0) mg per assay (approx. 0.10% of the sand in each assay). Accordingly, 7.9 mg was selected as a cutoff value, and bioassay trials in which the total sand displacement was equal to or less than this value were eliminated from analyses. One bioassay including 20 worker ants/assay (described below) fell below this threshold (workers escaped from arena) and was omitted from analysis. Five of the 96 bioassays involving six worker ants/assay (described below) failed to meet this threshold and the digging data were, therefore, omitted from analyses.

### 2.6. Behavioral Responses of Worker Ants to Bacterial Isolates

Binary choice bioassays were conducted to compare RIFA worker choice of digging site and digging activity between sand treated with bacterial isolates (17 individual isolates) and control sand. For each assay, treatment isolates were re-suspended to an approximate density of 5 × 10^8^ bacterial cells/mL, and 20 randomly chosen ants were placed into each arena. Two replicates from each of the eight colonies (four monogyne and four polygyne) were used to test worker ant responses to each bacterial isolate. Assays were conducted for 18 h and at the termination of each assay, the locations of ants were recorded, sand displaced from the treatment and control tubes was measured by weighing the microtubes, and DPIs were calculated. Because the DPIs normalized differences in the quantities of sand excavated by monogyne and polygyne ants, DPI data from this first series of assays (in which all 17 bacterial isolates were tested) was analyzed independently of RIFA social form to provide a generalized measure of RIFA worker responses to the bacterial isolate treatments. Accordingly, the three isolates with the highest average DPI and the three isolates with the lowest average DPI were selected for evaluation in additional assays in which bacterial isolates were presented at lower densities (approximately 5 × 10^6^ and 5 × 10^7^ cells/mL). Bioassays were performed in the manner described above (20 worker ants/test, 18 h).

### 2.7. Time-Course Observations

We conducted an additional series of bioassays in which the number of worker ants per assay was reduced. When using 20 ants per assay, recording the locations of individual ants was occasionally problematic. Additionally, ant locations were only determined at the termination of each assay, which provided only end-point observations of behavior. Using fewer ants allowed us to record the instantaneous locations of all ants over time. Based on preliminary assays (see Appendix A), we determined that six RIFA workers were the minimum number of ants required for a measurable amount of excavated sand (see Appendix A); assays were, therefore, performed with six ant workers per assay. Six bacterial isolates were evaluated with each administered to the sand at approximately 5 × 10^8^ cells/mL. The location of ants (treatment or control sand tubes) was recorded at 15, 30, 45, 60, 120, and 180 min following RIFA introduction. Finally, at 18 h, ant locations were again recorded, and as before, the weight of sand displaced, and DPI were calculated. For each ant social form, eight replicates of the bioassay were performed for each bacterial isolate.

### 2.8. Statistical Analyses

Statistical tests were performed using JMP Pro software (ver. 14.1.0, SAS Institute Inc., Cary, NC, USA). Shapiro–Wilk tests of bioassay data (ant location, excavated sand quantities, and DPI) showed that each followed a normal distribution. Student’s *t*-tests and analysis of variance (ANOVA) were therefore used to test these outcomes for effects of individual bacterial isolates. All tests were performed at *α* = 0.05. Paired *t*-tests were used to test for effects of bacterial isolates on RIFA locations and quantities of sand excavated for each of the bacterial isolates. ANOVA was used to test for effects of bacterial isolate on DPI. Post-hoc Tukey’s HSD tests were used to separate mean DPI values of the test groups. A repeated measures MANOVA (*α* = 0.05) was performed to test for the effects of colony social form and time on RIFA worker choice. Worker ant response was defined as the number of ants in each cohort responding at each time point (15, 30, 45, 60, 120, 180, and 1080 min) and served as the dependent variable in the repeated measures analysis.

## 3. Results

### 3.1. Bacteria Isolated from Ant Nest Soil

Colonies of purified bacterial isolates were identified by sequencing a fragment (600 base pairs) of the 16S rRNA gene. Seventeen isolates (Table 1) were selected and used in behavioral assays, including members of Firmicutes (*n* = 13), Actinobacteria (*n* = 3), and Proteobacteria (*n* = 1). Phylogenetic relationships among the selected bacteria are illustrated in Figure 2. The 16S rRNA sequences of the bacterial isolates were deposited in GenBank (Accession numbers MW255490 to MW255506).

### 3.2. Behavioral Responses of RIFA Workers to Bacterial Isolates

The initial series of tests generally showed strong effects of bacteria (administered at approx. 5 × 10^8^ cells/mL) on worker ant digging activity (at 18 h). Of the 17 isolates tested, 11 isolates had significant effects on worker ant choice of treatment or control tubes (Table 2) and 15 had significant effects on digging activity (Table 3). Based on digging activity, 13 bacterial isolates repelled RIFA workers; and based on digging site preference, 10 isolates repelled RIFA workers. Attraction of worker ants was only observed for sand treated with *Arthrobacter woluwensis* (paired *t*-test, *t*_(15)_ = 2.63, *p* = 0.010) and this isolate also resulted in significant digging activity (paired *t*-test, *t*_(15)_ = 2.36, *p* = 0.016). Attraction/repellency outcomes were generally consistent between digging site choice and digging activity parameters. Bacteria that repelled worker ants typically resulted in negative values for both of these variables, while the one bacterial isolate that was found to have significant effects on the attraction of worker ants had positive values for both of these variables.

Based on the average DPI calculated for each bacterial isolate (Table 4), isolates were generally repellent (having negative average DPI values). DPI values varied significantly between bacterial isolates (ANOVA, *F*_(17, 119)_ = 5.107, *p* < 0.0001) (Table 4). The most repellent isolates included *B. pacificus* (average DPI = −0.680), *Bacillus zanthoxyli* (average DPI = −0.657), and *Lysinibacillus pakistanensis* (average DPI = −0.545). *Arthrobacter woluwensis* was the only isolate that strongly attracted worker ants (average DPI = 0.205). Although no significant behavioral responses were elicited from isolates *Microbacterium xylanilyticum* (average DPI = 0.007) and *C. pusillum* (average DPI = −0.052), their DPIs were among the highest of the tested isolates, and *M. xylanilyticum* and *C. pusillum* were, therefore, regarded as bacteria that would potentially affect RIFA worker behavior when presented at different cell densities. Consequently, *B. pacificus*, *B. zanthoxyli*, *L. pakistanensis*, *A. woluwensis*, *M. xylanilyticum*, and *C. pusillum* were selected for further evaluations at lower cell densities.

### 3.3. Worker Ant Responses to Bacteria Isolates at Lower Cell Densities

The six isolates selected for further evaluation were tested at lower cell densities (approx. 5 × 10^7^ and 5 × 10^6^ cells/mL) in bioassays for monogyne and polygyne RIFA worker responses. Ant locations and digging activity (Table 5) responses varied among the bacterial isolates and cell densities tested. Results generally showed that most isolates repelled worker ants, though *A. woluwensis* attracted monogyne and polygyne RIFA workers. For most treatments, the site choice and digging outcomes agreed (both showing either negative or positive values for both parameters). On occasions in which the outcomes were contrary, (i.e., monogyne worker response to *M. xylanilyticum* (10^7^), polygyne worker response to *B. pacificus* (10^6^) and *L. pakistanensis* (10^6^), the responses were generally close to neutral and did not show strong attraction or repellency of worker ants to the bacteria.

At 5 × 10^7^ cells/mL (Table 5), *A. woluwensis* was the only isolate that attracted RIFA workers. Monogyne worker ant location and digging activity showed significant (paired *t*-tests, *p* < 0.05) preferences for *A. woluwensis*-treated sand over control sand. For polygyne worker ants, only their location outcome showed significant preference (paired *t*-test, *t*_(7)_ = 2.14, *p* = 0.035) for sand treated with *A. woluwensis* at this bacteria density. Bacterial isolates (5 × 10^7^ cell/mL) had significant effects on DPI (ANOVA, *F*_(5, 89)_ = 4.329, *p* < 0.0001); post-hoc analyses (Table 5) showed the average DPI of monogyne worker ants presented with *A. woluwensis* to be significantly higher (*p* < 0.05) than the DPI for monogyne and polygyne worker ants for all other isolates (except monogyne ants presented with *M. xylanilyticum*). At 5 × 10^7^ cells/mL, ant digging and site preference were significantly higher in control sand compared to sand treated with *B. pacificus* and *C. pusillum* (paired *t*-tests, *p* < 0.05) for both monogyne and polygyne ants. Monogyne worker ants preferred to dig in control sand (paired *t*-test, *t*_(7)_ = −6.98, *p* < 0.001) compared to *B. zanthoxyli* treatments, and the location of polygyne ants was significantly higher in control sand (paired *t*-test, *t*_(7)_ = −2.08, *p* = 0.038) compared to *B. zanthoxyli* treatments. Polygyne ants had significantly higher digging (paired *t*-test, *t*_(7)_ = −3.46, *p* = 0.005) and site choice (paired *t*-test, *t*_(7)_ = −1.90, *p* = 0.050) in control sand compared to sand treated with *L. pakistanensis*.

When bacterial isolates were presented at 5 × 10^6^ cells/mL (Table 6), *A. woluwensis* was again the only isolate that attracted RIFA workers. Significantly higher (paired *t*-test, *t*_(7)_ = 3.51, *p* = 0.005) numbers of polygyne ants were recorded digging in sand treated with *A. woluwensis* compared to the control sand. At this low bacterial density, ant digging results showed that *B. pacificus* (paired *t*-test, *t*_(7)_ = −2.38, *p* = 0.025) and *B. zanthoxyli* (paired *t*-test, *t*_(7)_ = −2.65, *p* = 0.016) repelled monogyne worker ants and *M. xylanilyticum* repelled polygyne worker ants (paired *t*-test, *t*_(7)_ = −1.94, *p* = 0.047). Results of binary assays showed that *B. pacificus* repelled monogyne workers (*p* = 0.029). Ant locations at 18 h, showed that significantly more monogyne ants preferred control sand over *C. pusillum*- (paired *t*-test, *t*_(7)_ = −2.54, *p* = 0.019) and *M. xylanilyticum*-treated sand (paired *t*-test, *t*_(7)_ = −2.34, *p* = 0.026), and polygyne ants preferred control sand over *L. pakistanensis*-treated sand (paired *t*-test, *t*_(7)_ = −2.42, *p* = 0.023). *C. pusillum* (10^6^ cells/mL) treated sand had significantly higher numbers of polygyne workers (paired *t*-test, *t*_(7)_ = 3.52, *p* = 0.005) than control sand. At 5 × 10^6^ cells/mL, bacterial isolates did not have a significant effect on monogyne and polygyne ant DPI (ANOVA: *F*_(5, 90)_ = 1.370, *p* = 0.2025) (Table 6).

Incidents of mortality of monogyne and polygyne ants in assays with 20 worker ants/assay were infrequent; the average number of dead ants per assay (±SE) was 1.14 (±0.15) ants at 5 × 10^8^ cells/mL, 0.72 (±0.15) ants at 5 × 10^7^ cells/mL, and 0.96 (±0.09) ants at 5 × 10^6^ cells/mL.

### 3.4. Time-Course of Worker Ant Responses to Bacterial Isolates

A time-course of observations was carried out so that the site selection and digging behavior of RIFA workers following introduction into the assay arena could be recorded. Monogyne RIFA workers exhibited significantly (paired *t*-test, *p* < 0.05) higher digging activity in *A. woluwensis* treated sand, but in contrast *Bacillus zanthoxyli* and *M. xylanilyticum* treatments elicited significantly higher digging responses in control sand (Table 7). Monogyne worker ants were repelled by *M. xylanilyticum* and *L. pakistanensis* treatments but attracted to the *A. woluwensis* treatment at the 180- and 1080-min time points (paired *t*-test, *p* < 0.05) (Figure 3). 

Polygyne RIFA worker ants were repelled by *L. pakistanensis* treatments from the 45 min to 180 min time points (paired *t*-test, *p* < 0.05) but were found in equivalent numbers (*p* > 0.05) in control and *L. pakistanensis*-treated sand at the terminal time point (Figure 3). *C. pusillium*-treated sand elicited significantly higher digging behavior (paired *t*-test, *t*_(7)_ = 2.024 *p* = 0.042) than control sand. Polygyne worker ants were attracted to *A. woluwensis*-treated sand at the terminal time point (paired *t*-test, *t*_(7)_ = −3.101 *p* = 0.009) but were observed in control sand in significantly higher numbers at 30 and 45 min.

During the early time points (15, 30, and 45 min), monogyne and polygyne RIFA worker ants explored treatment and control sands, but some ants remained in the neutral areas of the arenas. A significant difference in the choice of treatment or control sand did not typically occur until later time points (60, 120, 180, and 1080 min) (Figure 4). Colony social form had significant effects on worker ant responses (repeated measures MANOVA, *F*_(1)_ = 14.611, *p* = 0.0002) with polygyne ants having lower responder proportions compared to monogyne ants. Time was also found to be a significant factor (repeated measures MANOVA, *F*_(6)_ = 94.871, *p* < 0.0001); the proportions of RIFAs responding increased over the course of the bioassays.

In the time-course observations, the average (±SE) mortality of ants in assays conducted with six ants/trial was 0.28 (±0.05) ants.

## 4. Discussion

Previous research has demonstrated that RIFA workers were more attracted to nest-associated soils than to other soils, and that the Actinobacteria *Streptomyces* and *Nocardiopsis* produced volatile semiochemicals attractive to RIFA [31]. Our research followed up on these findings; we adopted a bioassay-guided approach to identify bacteria that attract or repel RIFAs. This approach required the cultivation of bacterial isolates for use in behavioral assays. We demonstrated that the choice of digging site and digging behaviors of *S. invicta* were significantly influenced by cultured bacterial isolates, with attraction to and avoidance of bacteria being dependent on the bacterial species and cell density. Most of the culturable bacteria that we isolated from RIFA nest soil were members of Firmicutes, including several Bacillaceae and Paenibacillaceae, which likely represented a minor portion of cultivable bacterial diversity, and are similar to the bacteria fauna previously cultured from *S*. *invicta* nests [43]. At the highest cell density tested, all of the Bacillaceae and Paenibacillaceae repelled RIFA worker ants. Several species of *Bacillus* are known pathogens of insects [44], and *Paenibacillus larvae* is a well-known pathogen of honeybees that causes American foulbrood [45,46]. RIFAs are susceptible to toxins produced by some strains of *B. thuringiensis* [47]. RIFA’s avoidance of these bacteria (in the concentrations that we tested), therefore, may be an adaptation to avoid potential infection. One of the isolates that we tested, *B. pacificus*, consistently repelled monogyne and polygyne RIFA workers when administered at various cell densities and ant cohort sizes.

In contrast to the Firmicutes, the three Actinobacteria that we evaluated did not strongly repel RIFA workers. Our results generally showed that *C. pusillum* and *M. xylanilyticum* elicited a moderately repellent to neutral response. *Arthrobacter woluwensis* was the only isolate that consistently attracted worker ants and elicited digging behavior in sand treated with this bacterium. RIFAs have been shown to be attracted to volatile emissions of Actinobacteria [31]. *Arthrobacter* species are aerobic bacteria widely distributed in the environment [48] and commonly found in soils [49]. *Arthrobacter* has been reported as a minor taxon present in RIFA’s gut microbiome [50,51]. In addition, Travanty et al. [32] identified higher abundances of Actinobacteria in RIFA nest soils than in adjacent uncolonized soil, which may be indicative of their association with the RIFA. Interactions of these bacteria and RIFA are, however, poorly understood. Interestingly, most reported associations with insects describe various *Arthrobacter* species as potential or confirmed pathogens of beetles [52,53]. Nevertheless, *Arthrobacter* has also been described as a component of the pine engraver beetle’s gut microbiome where it is thought to inhibit the growth of entomopathogenic fungi [54], and *Arthrobacter* species isolated from soil have been reported to inhibit fungal and bacterial development [55]. The inclination of RIFA workers to nest in *Arthrobacter*-treated sand suggests that these bacteria may convey protection to ants and brood by inhibiting microbial development in their nest. Huang et al. [31] showed that Actinobacteria derived from RIFA nest soil inhibited entomopathogenic fungi development. 

At lower bacterial cell densities, effects on RIFA worker behavior (attraction and repellency) decreased, confirming that these bacteria in soil affected ant behavior. As bacterial density decreased, the concentration of bacterial metabolites would also be expected to decrease. The dose-dependent results were similar to what has been described in other insect–bacteria interactions [56]. Identification of the specific bacterial metabolites that elicit RIFA repellency and attraction would allow dose-response experiments to be carried out [57,58]. Chemical identification of repellent metabolites might lead to the development of repellent compounds to protect electrical and other equipment from damage by RIFA.

It is notable that all of the bacteria tested in the current study were isolated from RIFA nest soil, suggesting that RIFAs are tolerant of their presence in natural concentrations. We did not perform an analysis of the bacterial densities of native soils; consequently, it is not clear at what densities these bacteria normally occur. The current study shows that bacteria cultured from RIFA nest soils affects worker ant digging, though the bacterial communities of RIFA nest soils are highly diverse [32] and include many non-culturable species that may affect RIFA worker digging activities. Although culture-dependent methods generally recover a small portion of the diversity from soil environments, culturable bacteria are critical for research of insect–microbe interactions [59]. For example, bacteria have been shown to produce several volatile compounds (including alcohols, acids, ketones, pyrazines, and phenols) that attract insects [27,59], including RIFA [31]. Therefore, we used culturable bacteria for behavioral studies with RIFA.

Time course evaluations generally showed that monogyne RIFA worker ants responded to bacterial isolates more rapidly than polygyne RIFA worker ants. These results suggested that polygyne workers may be less sensitive to bacteria and bacterial metabolites than monogyne. These results are consistent with RIFA social phenotype in that polygyne RIFA exhibit a reduced ability to discriminate between nestmates [60]. Genetically, monogyne and polygyne RIFA workers differ in their genotype of the social chromosome—monogyne workers have only the SB/SB genotype, whereas polygyne workers are a 50:50 mix of SB/Sb and SB/SB genotypes. Transcriptome analyses have shown differential gene expression between *S. invicta* SB/Sb workers and monogyne and polygyne workers, including the underexpression of several proteins related to chemoreception in SB/Sb RIFA workers which only occur in polygyne colonies [9]. This genetic difference may also contribute to the variation between monogyne and polygyne RIFAs in their sensitivity to environmental microbial emissions.

During the time-course experiment, RIFA workers were generally observed to make contact with both treatment and control sands prior to making a choice of where to reside and focus their digging activity. Typically, worker ants would visit both options and then continue to explore neutral areas before taking refuge. The response of polygyne RIFA worker ants to *Paenibacillus* sp. was notable, in that worker ants had a strong aversion to these bacteria for the first 3 h, but at the termination of the assay, ants were found in treatment and control tubes in equal proportions. In this case, it is possible that the bacterial metabolites dissipated over the course of the experiment, and after 18 h, bacteria-treated sand lacked sufficient repellents to affect the RIFA. To distinguish between these possibilities, future studies may be designed with conditions that support bacterial viability through the duration of RIFA exposure, or assays could be performed over a shorter time-course, thus ensuring that RIFAs are continuously exposed to bacterial metabolites.

## 5. Conclusions

Our findings indicate that soil bacteria can affect RIFA worker digging behavior and the behavioral responses that we observed were dependent on the bacterial species tested. Responses of RIFA workers ranged from strong aversion to attraction and were dependent on bacterial species and cell density. The chemical identification of bacterial metabolites that elicited behavioral reactions merit additional research. These compounds may provide a basis to develop novel management strategies that repel RIFA away from sensitive infrastructure or provide attraction to insecticidal baits.

## Figures and Tables

**Figure 1 insects-13-00444-f001:**
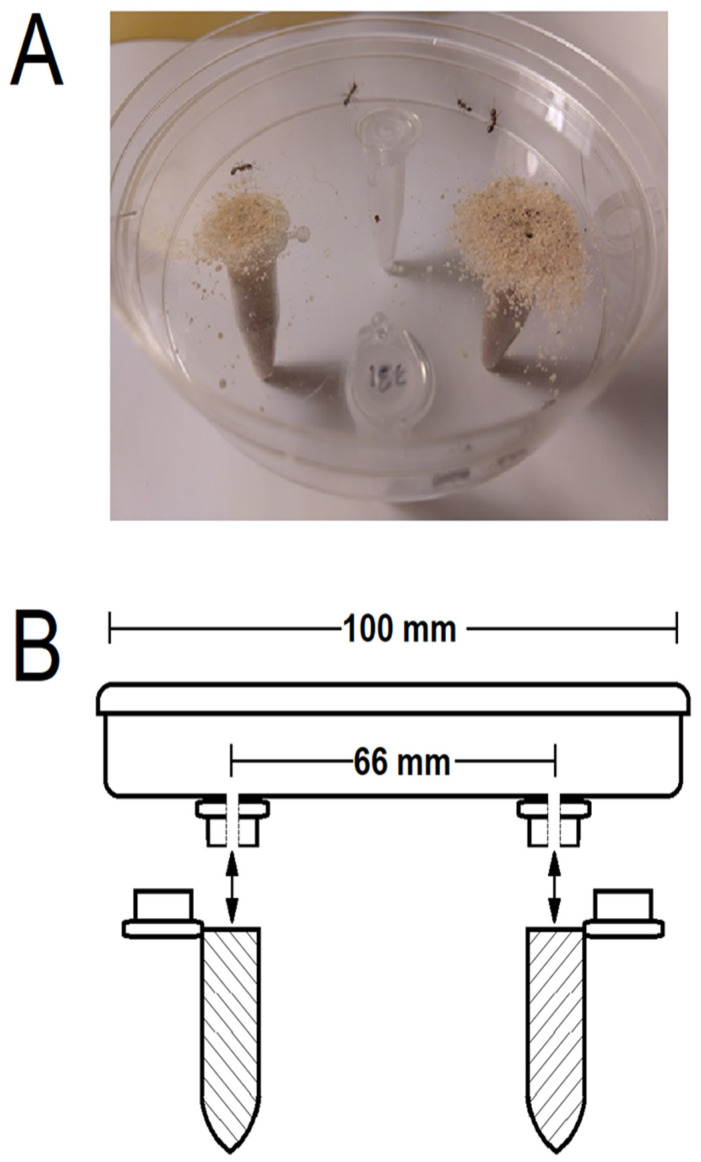
(**A**) A photograph of the bioassay apparatus with active RIFA workers present. The treatment (left) and control (right) sand vials are positioned on opposite sides of an arena. RIFA (*n* = 20 or 6 per assay, dependent on assay) excavate sand from available vials; excavated sand is quantified, and ant positions are determined to infer repellency/attraction of bacterial treatments. Two additional empty support vials are fixed to the Petri dish to provide stability to the apparatus. (**B**) A diagram of the bioassay apparatus. The caps of two 1.5 mL microcentrifuge tube caps are fixed to the bottom of a 100 mm petri dish; holes (3 mm) are spaced 66 mm apart through the caps and Petri dish and provide access to treatment and control sand tubes. Sand tubes are attached to apparatus prior to the assay and removed at the assay termination.

**Figure 2 insects-13-00444-f002:**
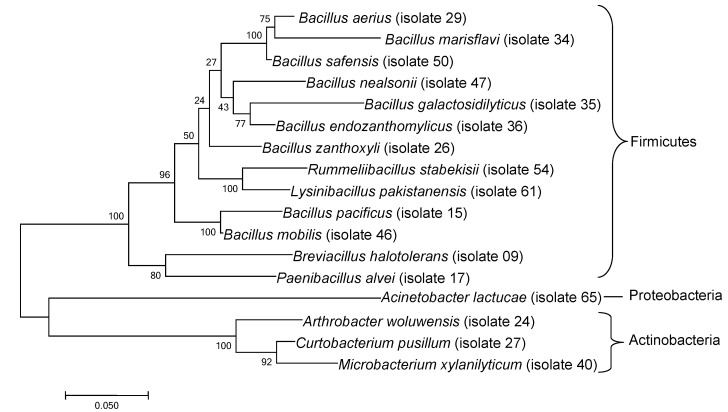
Phylogenetic tree reconstructed by the maximum likelihood method (Tamura-Nei model) based on 16S rRNA gene sequences showing the phylogenetic relationship between bacteria cultured from *Solenopsis invicta* nest soils. Bootstrap percentage values (based on 500 replications) with 50% cutoff value are shown at the nodes. Phylum-level groupings indicated on right.

**Figure 3 insects-13-00444-f003:**
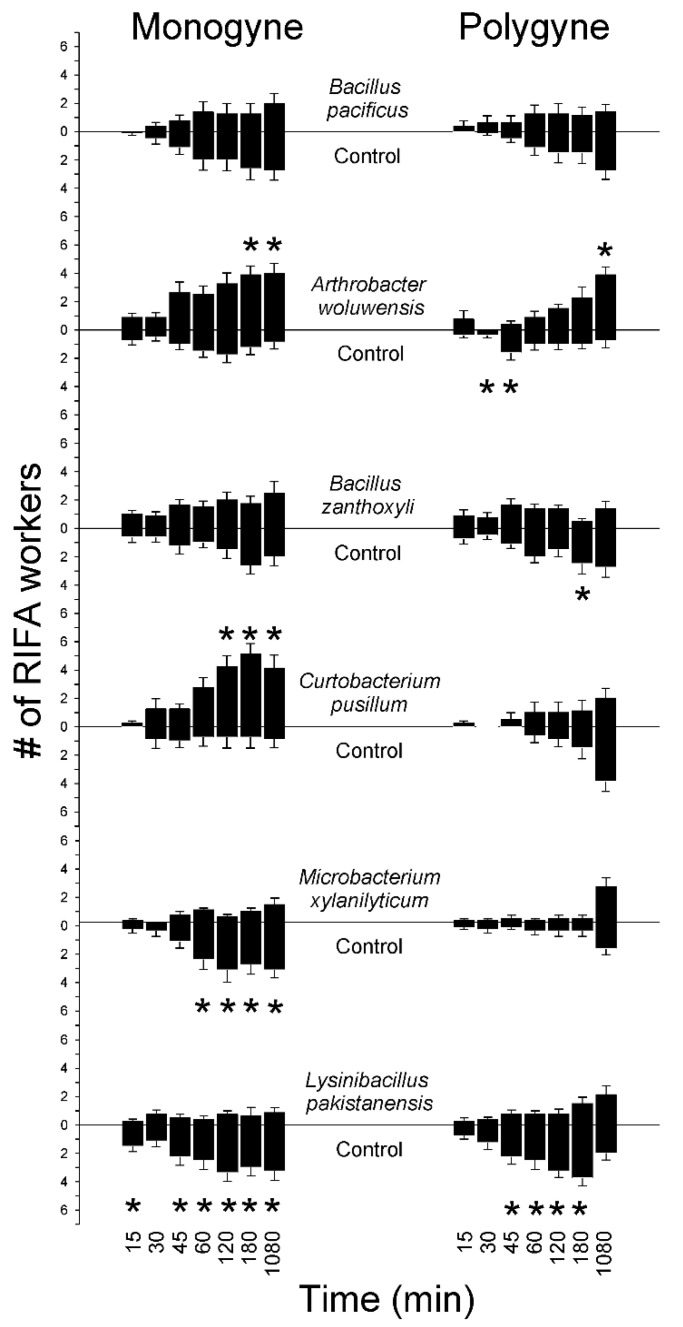
Location of monogyne and polygyne *S. invicta* in response to soil-derived bacterial isolates (5 × 10^8^ cells/mL) or control (untreated) sand in 2-choice bioassays. Location of responders measured at several time intervals (min), terminating at 18 h; 6 worker ants/trial, *n* = 8 trials. Asterisks indicate significant preference (paired *t*-tests, *α* = 0.05) of RIFA workers. #: number.

**Figure 4 insects-13-00444-f004:**
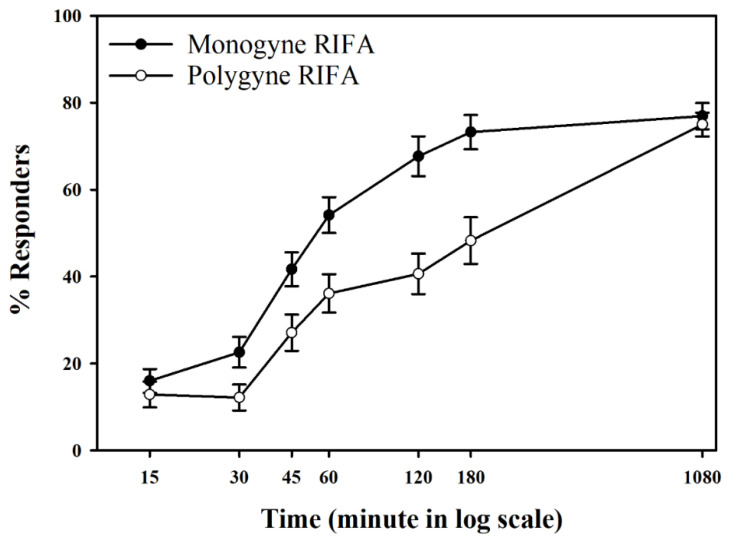
Monogyne and polygyne *S. invicta* responder percentages (±SE) in 2-choice bioassays at various time intervals. Each assay trial includes *n* = 6 worker ants; responders are located in/on treatment or control areas of bioassay arena; non-responders are in neutral areas (away from treatment and control).

**Table 1 insects-13-00444-t001:** Bacteria isolated from the *Solenopsis invicta* nest soil.

Closest Cultured Bacteria/Sequence from NCBI (Strain)	Classification (Phylum)	Similarity (%)	Closest Match NCBI Accession Number	Deposited in NCBI with Accession Number
*Brevibacillus halotolerans* (LAM0312)	Firmicutes	99.80	NR156834	MW255490
*Bacillus pacificus* (MCCC 1A06182)	Firmicutes	99.80	NR157733	MW255491
*Paenibacillus alvei* (NBRC 3343)	Firmicutes	99.20	NR113577	MW255492
*Arthrobacter woluwensis* (1551)	Actinobacteria	98.30	NR044894	MW255493
*Bacillus zanthoxyli* (1433)	Firmicutes	99.80	NR164882	MW255494
*Curtobacterium pusillum* (DSM 20527)	Actinobacteria	98.20	NR042315	MW255495
*Bacillus aerius* (24K)	Firmicutes	99.80	NR118439	MW255496
*Bacillus marisflavi* (TF-11)	Firmicutes	99.80	NR118437	MW255497
*Bacillus galactosidilyticus* (LMG 17892)	Firmicutes	96.70	NR025580	MW255498
*Bacillus endozanthoxylicus* (1404)	Firmicutes	96.20	NR158107	MW255499
*Microbacterium xylanilyticum* (S3-E)	Actinobacteria	98.20	NR042350	MW255500
*Bacillus mobilis* (MCCC 1A05942)	Firmicutes	99.80	NR157731	MW255501
*Bacillus nealsonii* (DSM 15077)	Firmicutes	98.50	NR044546	MW255502
*Bacillus safensis* (FO-36b)	Firmicutes	100	NR041794	MW255503
*Rummeliibacillus stabekisii* (NBRC 104870)	Firmicutes	99.80	NR114270	MW255504
*Lysinibacillus pakistanensis* (NCCP-54)	Firmicutes	99.80	NR113166	MW255505
*Acinetobacter lactucae* (NRRL B-41902)	Proteobacteria	99.30	NR152004	MW255506

**Table 2 insects-13-00444-t002:** Effects of soil-derived bacterial isolates on *S. invicta* residing preferences after 18 h.

Bacterial Isolate	Number of Ants (±SE)		% Ants Responding (±SE)	*t*-Test on Ant Choice *t*-Value (*P* > *t*)	*DF*
	Treatment	Control			
*B. halotolerans*	8.1 (0.6)	8.5 (0.8)	82.6 (3.5)	−0.29 (0.387)	15
*B. pacificus*	3.4 (1.1)	13.8 (1.2)	84.2 (2.9)	**−4.70 (<0.001)**	15
*P. alvei*	3.1 (1.4)	13.1 (1.7)	80.6 (6.1)	**−3.45 (0.002)**	15
*A. woluwensis*	9.3 (0.9)	5.0 (0.9)	71.0 (4.2)	**2.63 (0.010)**	15
*B. zanthoxyli*	1.6 (0.6)	13.4 (1.1)	74.5 (5.6)	**−8.98 (<0.001)**	15
*C. pusillum*	8.8 (1.3)	6.5 (1.2)	75.8 (5.0)	0.97 (0.174)	15
*B. aerius*	3.3 (0.7)	8.33 (1.5)	58.9 (7.6)	**−2.85 (0.006)**	15
*B. marisflavi*	1.9 (0.8)	14.4 (1.0)	81.5 (3.5)	**−7.54 (<0.001)**	15
*B. galactosidilyticus*	6.3 (1.0)	8.6 (1.0)	74.2 (3.7)	−1.24 (0.116)	15
*B. endozanthoxylicus*	3.9 (1.3)	11.8 (1.4)	78.4 (3.4)	**−3.07 (0.004)**	15
*M. xylanilyticum*	7.9 (0.9)	8.3 (1.1)	80.6 (3.0)	−0.20 (0.424)	15
*B. mobilis*	2.3 (0.7)	12.1 (1.2)	73.0 (4.3)	**−5.88 (<0.001)**	15
*B. nealsonii*	5.8 (1.1)	10.2 (1.3)	79.4 (4.2)	**−1.95 (0.035)**	15
*B. safensis*	7.4 (1.2)	7.2 (1.1)	73.2 (5.3)	0.09 (0.464)	15
*R. stabekisii*	4.2 (1.2)	11.2 (1.6)	77.1 (5.2)	**−2.63 (0.010)**	15
*L. pakistanensis*	7.3 (1.8)	8.4 (1.8)	78.4 (3.1)	−0.30 (0.385)	15
*A. lactucae*	3.7 (0.7)	10.1 (1.0)	73.3 (4.2)	**−4.83 (<0.001)**	15

Bacterial suspensions (0.35 mL) presented at 5 × 10^8^ cells/mL were applied to approx. 3 g soil. Paired *t*-tests (*α* = 0.05) compare locations of RIFA for 16 trials (8 monogyne + 8 polygyne), with 20 RIFA workers/trial. Negative *t*-values indicate repellent response and positive *t*-values indicate attractive response by RIFA; significant results shown in bold.

**Table 3 insects-13-00444-t003:** Effects of soil-derived bacterial isolates on *S. invicta* digging preferences after 18 h.

Bacterial Isolate	Sand Removed (mg) (±SE)		*t*-Test on Sand Removed *t*-Value (*P* > *t*)	*DF*
	Treatment	Control		
*B. halotolerans*	775.3 (62.0)	998.4 (66.7)	**−4.09 (<0.001)**	15
*B. pacificus*	123.4 (48.2)	430.9 (82.0)	**−4.56 (<0.001)**	15
*P. alvei*	144.4 (56.3)	332.8 (65.9)	**−1.97 (0.034)**	15
*A. woluwensis*	531.0 (86.6)	351.0 (53.2)	**2.36 (0.016)**	15
*B. zanthoxyli*	205.4 (71.5)	667.4 (57.7)	**−9.04 (<0.001)**	15
*C. pusillum*	634.7 (69.1)	783.6 (99.0)	−1.19 (0.127)	15
*B. aerius*	71.8 (17.6)	184.1 (47.9)	**−2.24 (0.021)**	15
*B. marisflavi*	136.8 (49.3)	347.0 (61.8)	**−3.65 (0.001)**	15
*B. galactosidilyticus*	376.7 (65.4)	501.6 (66.7)	**−2.32 (0.017)**	15
*B. endozanthoxylicus*	253.1 (69.5)	472.9 (83.1)	**−2.95 (0.005)**	15
*M. xylanilyticum*	577.0 (68.5)	556.4 (58.8)	0.40 (0.346)	15
*B. mobilis*	267.6 (77.0)	509.6 (70.0)	**−4.26 (<0.001)**	15
*B. nealsonii*	415.8 (92.7)	559.8 (76.9)	**−2.17 (0.023)**	15
*B. safensis*	956.2 (71.2)	1124.4 (61.6)	**−2.77 (0.007)**	15
*R. stabekisii*	351.3 (63.2)	615.6 (66.9)	**−2.23 (0.021)**	15
*L. pakistanensis*	312.6 (59.7)	880.3 (65.6)	**−8.63 (<0.001)**	15
*A. lactucae*	337.6 (65.6)	535.4 (80.5)	**−2.70 (0.008)**	15

Bacterial suspensions (0.35 mL) presented at 5 × 10^8^ cells/mL were applied to approx. 3 g soil. Paired *t*-tests (*α* = 0.05) compare quantities of sand removed for 16 trials (8 monogyne + 8 polygyne), with 20 RIFA workers/trial. Negative *t*-values indicate repellent response and positive *t*-values indicate attractive response by RIFA; significant results shown in bold.

**Table 4 insects-13-00444-t004:** Effects of soil-derived bacterial isolates (5 × 10^8^ cells/mL) on *S. invicta* digging preference index (DPI) after 18 h.

Bacterial Isolate	Average DPI (±SE)	Tukey’s Test to Identify Significantly Different Means			
*A. woluwensis*	0.21 (0.11)	A			
*M. xylanilyticum*	0.01 (0.05)	A	B		
*C. pusillum*	−0.05 (0.11)	A	B	C	
*B. safensis*	−0.09 (0.03)	A	B	C	
*B. halotolerans*	−0.13 (0.03)	A	B	C	
*B. galactosidilyticus*	−0.20 (0.08)	A	B	C	
*B. mobilis*	−0.22 (0.12)	A	B	C	D
*R. stabekisii*	−0.27 (0.14)	A	B	C	D
*B. aerius*	−0.28 (0.13)	A	B	C	D
*A. lactucae*	−0.28 (0.10)	A	B	C	D
*B. endozanthoxylicus*	−0.34 (0.14)		B	C	D
*P. alvei*	−0.44 (0.19)		B	C	D
*B. mobilis*	−0.45 (0.10)		B	C	D
*B. marisflavi*	−0.52 (0.12)			C	D
*L. pakistanensis*	−0.55 (0.07)			C	D
*B. zanthoxyli*	−0.66 (0.08)				D
*B. pacificus*	−0.68 (0.09)				D

Bioassays (20 worker ants/trial) were conducted with monogyne (*n* = 8 trials) and polygyne (*n* = 8 trials) ants for a total of *n* = 16 trials/isolate. There were significant differences in the responses to different isolates (ANOVA: *F* = 5.107, *p* < 0.0001). DPIs that do not share letters (A, B, C, and D) are significantly different from each other (Tukey’s HSD test, *p* < 0.05). DPI values ranged from −1.0 (indicating relative preference for control sand) to 1.0 (indicating relative preference for treated sand).

**Table 5 insects-13-00444-t005:** Effects of soil-derived bacterial isolates presented at 5 × 10^7^ cells/mL densities on monogyne and polygyne *S. invicta* location and digging activity in bacteria-treated or control sand and digging preference index (DPI) after 18 h.

Bacterial Isolate	Ant Social Form †	Number of Ants (±SE)		Mean Amount of Sand (mg) Excavated (±SE)		Average DPI (±SE) ‡	*t*-Test on Ant Choice *t*-Value (*P*) §	*t*-test on Ant Digging Activity *t*-Value (*P > t*) §	*DF*
		Treatment	Control	Treatment	Control				
*A. woluwensis*	M	16.1 (1.1)	3.0 (0.9)	477.3 (92.9)	209.0 (70.3)	0.53 (0.13) ^a^	**6.86 (<0.001)**	**3.56 (0.005)**	7
	P	11.8 (1.8)	5.4 (1.3)	293.9 (95.5)	145.8 (32.1)	0.13 (0.19) ^a,b^	**2.14 (0.035)**	1.82 (0.056)	7
*M. xylanilyticum*	M	8.9 (2.3)	8.5 (2.1)	744.0 (166.5)	798.8 (103.1)	−0.10 (0.18) ^a,b,c^	0.09 (0.467)	−0.22 (0.415)	7
	P	7.5 (1.6)	8.6 (1.8)	532.0 (146.5)	589.5 (69.9)	−0.19 (0.18) ^b,c^	−0.37 (0.363)	−0.45 (0.334)	7
*C. pusillum*	M	1.9 (0.8)	14.3 (1.7)	561.5 (108.2)	1084.8 (67.8)	−0.24 (0.11) ^b,c^	**−4.90 (<0.001)**	**−4.05 (0.002)**	7
	P	4.1 (1.1)	8.0 (0.5)	485.8 (90.3)	713.1 (81.0)	−0.36 (0.11) ^b,c^	**−2.83 (0.013)**	**−3.28 (0.007)**	7
*L. pakistanensis*	M	7.4 (2.0)	8.6 (1.4)	103.7 (78.4)	149.0 (55.4)	−0.37 (0.17) ^b,c^	−0.32 (0.365)	−0.91 (0.198)	6
	P	6.0 (1.3)	10.0 (0.9)	115.9 (36.0)	207.4 (45.8)	−0.37 (0.07) ^b,c^	**−1.90 (0.050)**	**−3.46 (0.005)**	7
*B. zanthoxyli*	M	4.5 (2.2)	12.0 (2.0)	553.8 (96.6)	1120.5 (49.6)	−0.38 (0.15) ^b,c^	−1.81 (0.057)	**−6.98 (<0.001)**	7
	P	3.6 (1.5)	10.5 (2.1)	328.3 (115.0)	595.1 (91.1)	−0.39 (0.11) ^b,c^	**−2.08 (0.038)**	−1.78 (0.059)	7
*B. pacificus*	M	3.0 (0.6)	13.9 (1.3)	480.3 (140.8)	904.3 (113.4)	−0.46 (0.15) ^b,c^	**−5.89 (<0.001)**	**−2.97 (0.010)**	7
	P	1.1 (0.4)	15.0 (1.0)	56.9 (17.2)	562.9 (138.0)	−0.65 (0.18) ^c^	**−11.84 (<0.001)**	**−3.84 (0.003)**	7

† M = monogyne, P = Polygyne. ‡ Soil-derived bacterial isolates presented at 5 × 10^7^ cells/mL have significant effects on monogyne and polygyne RIFA digging preference index (DPI) after 18 h (ANOVA: *F* = 4.329, *p* < 0.0001). DPIs that do not share letters (a, b, and c) are significantly different from each other (Tukey’s HSD test, *p* < 0.05). DPI values ranged from −1.0 (indicating relative preference for control sand) to 1.0 (indicating relative preference for treated sand). § Paired *t*-tests (*α* = 0.05) compare outcomes with 20 worker ants/trial. Negative *t*-values indicate repellency, positive *t*-values indicate attraction; significant results shown in bold.

**Table 6 insects-13-00444-t006:** Effects of soil-derived bacterial isolates presented at 5 × 10^6^ cells/mL densities on monogyne and polygyne *S. invicta* location and digging activity in bacteria-treated or control sand and digging preference index (DPI) after 18 h.

Bacterial Isolate	Ant Social Form †	Number of Ants (±SE)		Mean Amount of Sand (mg) Excavated (±SE)		Average DPI (±SE) ‡	*t*-test on Ant Choice *t*-Value (*P*) §	*t*-Test on ant Digging Activity *t*-Value (*P > t*) §	*DF*
		Treatment	Control	Treatment	Control				
*A. woluwensis*	M	8.9 (2.2)	9.3 (2.6)	535.5 (93.5)	560.9 (135.5)	0.29 (0.19)	−0.08 (0.470)	−0.22 (0.418)	7
	P	12.6 (1.2)	4.4 (1.4)	349.5 (105.0)	252.4 (82.6)	0.11 (0.12)	**3.51 (0.005)**	0.85 (0.212)	7
*M. xylanilyticum*	M	4.4 (1.1)	10.5 (1.6)	77.4 (14.2)	144.3 (35.7)	0.06 (0.06)	**−2.34 (0.026)**	−1.53 (0.085)	7
	P	1.8 (0.6)	3.6 (0.7)	12.9 (1.7)	56.4 (22.6)	0.02 (0.12)	−1.64 (0.072)	**−1.94 (0.047)**	7
*C. pusillum*	M	5.3 (1.0)	9.6 (0.9)	626.4 (115.9)	658.9 (145.4)	−0.01 (0.10)	**−2.54 (0.019)**	−0.35 (0.368)	7
	P	8.6 (0.7)	3.5 (1.0)	371.1 (31.9)	342.4 (49.1)	−0.14 (0.12)	**3.52 (0.005)**	0.7 (0.252)	7
*L. pakistanensis*	M	7.0 (0.8)	9.3 (0.6)	900.9 (56.6)	810.6 (138.0)	−0.19 (0.33)	−1.76 (0.061)	0.62 (0.279)	7
	P	3.9 (1.2)	9.0 (1.1)	247.3 (50.7)	303.4 (37.7)	−0.21 (0.21)	**−2.42 (0.023)**	0.84 (0.214)	7
*B. zanthoxyli*	M	4.3 (1.6)	10.4 (2.3)	372.8 (130.0)	559.5 (132.9)	−0.25 (0.11)	−1.61 (0.076)	**−2.65 (0.016)**	7
	P	5.8 (2.3)	10.0 (2.5)	124.6 (62.6)	152.8 (44.0)	−0.27 (0.19)	−0.92 (0.195)	−0.31 (0.383)	7
*B. pacificus*	M	2.8 (1)	10.0 (2.4)	63.4 (44.8)	216.1 (85.5)	−0.37 (0.27)	**−2.27 (0.029)**	**−2.38 (0.025)**	7
	P	9.5 (3.2)	6.9 (2.8)	194.3 (106.6)	245.3 (84.8)	−0.42 (0.15)	0.47 (0.327)	−0.74 (0.242)	7

† M = monogyne, P = Polygyne. ‡ Soil-derived bacterial isolates presented at 5 × 10^6^ cell/mL do not have significant effects on monogyne and polygyne RIFA digging preference index (ANOVA: *F* = 1.370, *p* <0.2025). DPI values ranged from −1.0 (indicating relative preference for control sand) to 1.0 (indicating relative preference for treated sand). § Paired *t*-tests (*α* = 0.05) compare outcomes with 20 worker ants/trial. Negative *t*-values indicate repellency, positive *t*-values indicate attraction; significant results shown in bold.

**Table 7 insects-13-00444-t007:** Effects of soil-derived bacterial isolates (5 × 10^8^ cells/mL) on *S. invicta* digging activity with 6 worker ants per assay after 18 h.

RIFA Social Form	Bacterial Isolate	Sand Removed (±SE) mg		*t*-Test on Amount of Sand Removed *t*-Value (*P* > *t*)	*DF*	DPI (±SE)
		Treatment	Control			
Monogyne	*B. pacificus*	21.3 (5.5)	41.4 (12.2)	−1.563 (0.081)	7	−0.197 (0.173)
	*A. woluwensis*	169.7 (62.2)	65.7 (29.7)	**2.126 (0.043)**	6	0.377 (0.194)
	*B. zanthoxyli*	34.1 (21.6)	107.6 (38.7)	**−2.735 (0.015)**	7	−0.342 (0.232)
	*C. pusillum*	211.5 (77.4)	82 (32.6)	1.432 (0.098)	7	0.137 (0.267)
	*M. xylanilyticum*	18.6 (7.6)	86.4 (32.1)	**−2.366 (0.025)**	7	−0.526 (0.139)
	*L. pakistanensis*	22.8 (11.9)	75.9 (37.6)	−1.214 (0.132)	7	−0.27 (0.288)
Polygyne	*B. pacificus*	6.9 (1.8)	8 (1.3)	−0.471 (0.327)	6	−0.136 (0.181)
	*A. woluwensis*	49.7 (36)	16.6 (2.7)	0.893 (0.203)	6	−0.09 (0.276)
	*B. zanthoxyli*	40.1 (21.5)	90.7 (62.6)	−1.193 (0.139)	6	−0.133 (0.239)
	*C. pusillum*	23 (5.4)	10 (3)	**2.024 (0.042)**	7	0.321 (0.174)
	*M. xylanilyticum*	9.4 (1.9)	30.5 (24.1)	−0.838 (0.218)	7	0.063 (0.194)
	*L. pakistanensis*	9.3 (3.3)	20.1 (5.9)	−1.748 (0.066)	6	−0.406 (0.123)

Paired *t*-tests (*α* = 0.05) compare quantities of removed sand for 16 trials (8 monogyne + 8 polygyne), with 6 worker ants/trial. DPI values ranged from −1.0 (indicating relative preference for control sand) to 1.0 (indicating relative preference for treated sand). Negative *t*-values indicate preference for control sand, positive *t*-values indicate preference for treatment sand; statistically significant results are shown in bold.

## Data Availability

Data is contained within the article and Appendix A.

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
