# Peer review of "Bacterial Isolates Derived from Nest Soil Affect the Attraction and Digging Behavior of Workers of the Red Imported Fire Ant, *Solenopsis invicta* Buren"

_insects, 2022, doi:10.3390/insects13050444_

Round 1

Reviewer 1 Report

See attached. Some of the methods and results need to be clarified.

Reviewer 2 Report

This nice study cultured 17 bacteria from RIFA nest soils. The effects of individual bacterial isolates and bacterial cell densities on residing preference and digging activity of monogyne and polygyne RIFA worker ants were evaluated. The effective bacterial isolates may have the potential to control this invasive pest. The experiments were well conducted, and the manuscript was professionally prepared.

Line 135: Normally, primers (27F and 1492R pair) were used to amplify bacterial 16S rRNA genes.

Line 154: Is a hemocytometer reliable method to measure bacterial densities? Are there any attempts to use other methods?

Line 155: Will the 0.85% NaCl affect the bacterial vitality?

Line 157: Is 96% similarity too low here?

Line 305: Why are there wide variation in sand removed by control ants? Is it due to size differences in workers selected?

For most tables and figures, please find a way to list bacterial isolates in a proper order to ease following by readers.
